# Paralytic Shellfish Toxins of *Pyrodinium bahamense* (Dinophyceae) in the Southeastern Gulf of Mexico

**DOI:** 10.3390/toxins14110760

**Published:** 2022-11-03

**Authors:** Erick J. Núñez-Vázquez, Carlos A. Poot-Delgado, Andrew D. Turner, Francisco E. Hernández-Sandoval, Yuri B. Okolodkov, Leyberth J. Fernández-Herrera, José J. Bustillos-Guzmán

**Affiliations:** 1Centro de Investigaciones Biológicas del Noroeste (CIBNOR), Apdo. Postal 128, La Paz 23000, Mexico; 2Investigación para la Conservación y el Desarrollo (INCODE), Nayarit 1325 A. Col. Las Garzas, La Paz 23079, Mexico; 3Tecnológico Nacional de México/Instituto Tecnológico Superior de Champotón, Campeche (TECNM-ITESCHAM), Carretera Champotón, Isla Aguada Km 2, Col. El Arenal, Champotón 4400, Mexico; 4Centre for Environment Fisheries and Aquaculture Science (CEFAS), Food Safety Group, Barrack Road, Weymouth DT4 8UB, UK; 5Instituto de Ciencias Marinas y Pesquerías (ICIMAP-UV), Universidad Veracruzana, Calle Mar Mediterráneo Núm. 314, Fracc. Costa Verde, Boca del Río 9429, Mexico

**Keywords:** paralytic shellfish toxins, paralytic shellfish poisoning, saxitoxin, neosaxitoxin, *Pyrodinium bahamense*, Gulf of Mexico, HPLC-FLD, UHPLC-MS/MS, phytoplankton, dinoflagellates

## Abstract

In September and November 2016, eight marine sampling sites along the coast of the southeastern Gulf of Mexico were monitored for the presence of lipophilic and hydrophilic toxins. Water temperature, salinity, hydrogen potential, dissolved oxygen saturation, inorganic nutrients and phytoplankton abundance were also determined. Two samples filtered through glass fiber filters were used for the extraction and analysis of paralytic shellfish toxins (PSTs) by lateral flow immunochromatography (IFL), HPLC with post-column oxidation and fluorescent detection (FLD) and UHPLC coupled to tandem mass spectrometry (UHPLC-MS/MS). Elevated nutrient contents were associated with the sites of rainwater discharge or those near anthropogenic activities. A predominance of the dinoflagellate *Pyrodinium bahamense* was found with abundances of up to 10^4^ cells L^−1^. Identification of the dinoflagellate was corroborated by light and scanning electron microscopy. Samples for toxins were positive by IFL, and the analogs NeoSTX and STX were identified and quantified by HPLC-FLD and UHPLC-MS/MS, with a total PST concentration of 6.5 pg cell^−1^. This study is the first report that confirms the presence of PSTs in *P. bahamense* in Mexican waters of the Gulf of Mexico.

## 1. Introduction

The dinoflagellate *Pyrodinium bahamense* Plate 1906 (Dinoflagellata: Dinophyceae: Goniodomaceae) is a tropical–subtropical species that was originally described from the Atlantic Ocean at New Providence Island (Bahamas) [1]. The first record of paralytic shellfish poisoning (PSP) caused by *P. bahamense* was recorded in 1972 in Southeast Asia at Port Moresby, Papua New Guinea [2,3]. It is one of the main dinoflagellate species producing paralytic shellfish toxins (PSTs) that have caused PSP from the consumption of bivalves, planktivorous fish and gastropods in the world [4,5,6,7,8], mainly in tropical and subtropical areas [7,9,10,11]. Their proliferations have also been associated with epizootics and massive mortalities in fish, sea turtles and other marine organisms [8,12,13,14,15,16,17,18,19,20]. Emslie et al. [21] suggested that mass mortality of the extinct cormorant *Phalacrocorax filyawi* Emslie, other seabird, fish and a seal species about 54 million years ago in coastal waters off Sarasota, Florida, United States, coincided with accumulation of *P. bahamense* cysts that have a stratigraphic range from the lower Eocene to the Holocene. The species was also described as a producer of the PSTs (saxitoxin and analogs) related to human intoxications due to consumption of puffer fish (Tetraodontidae) in Florida, in which, unexpectedly, tetrodotoxin and analogs were not detected as the cause of intoxication, thus establishing a new syndrome called saxitoxin puffer fish poisoning (SPFP) [22].

In the southeastern Mexican Pacific, *P. bahamense* has had the greatest impact on fisheries, tourism and coastal marine ecosystems due to the implementation of long sanitary bans, in which the extraction and commercialization of shellfish is prohibited [23]. Such incidents affect mainly bivalve harvesting with the purpose of protecting public health, because it has been the cause of 476 human poisonings and 18 deaths from PSP in the Mexican tropical Pacific [8,20,24,25,26,27,28,29]. In May 2022, a new case of PSP with 12 poisonings and two deaths was reported from the coastal waters of the state of Chiapas [30], and the health authorities of the Government of Chiapas issued a sanitary ban for the extraction and commercialization of bivalve mollusks [31].

Some authors distinguish two varieties or morphotypes in *P. bahamense* var. *compresum* and var. *bahamense*. Initially, var. *bahamense* was thought to be confined to the Atlantic Ocean and var. *compresum* to be limited to the Pacific Ocean [7,32]. However, differentiation between the two varieties based on morphological criteria is not irrefutable, as shown by a detailed morphological analysis of the populations from Papua, the Philippines, Jamaica and Puerto Rico [33]. In addition, the physiological criterion of toxin production versus no production is no longer applicable [11]. Therefore, segregated biogeography is no longer accepted because both varieties have been reported to coexist in several places, such as Costa Rica [34], the Pacific coast of Mexico [35] and along the eastern coast of Florida [22].

In the Gulf of Mexico and the Caribbean Sea, *P. bahamense* appears to have a continuous distribution [36,37,38,39], also being present in coastal lagoons [40,41], with a range of abundances of 10^3^–10^5^ cells L^−1^, generally in summer [38]. The species has been recorded along the central coast of Campeche (southwestern of Gulf de Mexico) with minimum abundances of 2 × 10^4^ cells L^−1^ recorded in October 2008 to a maximum of 3.3 × 10^5^ cells L^−1^ observed in August 2005 [41]. The first evidence of the presence of PSTs along the Campeche central coast with the profile of STX, NeoSTX, GTX2-3 and dcGTX2-3 was attributed to *P. bahamense*. This was inferred [42,43] indirectly from the examination of toxins in puffer fish *Sphoeroides* spp. and *Lagocephalus laevigatus* and thought to probably be transmitted via bivalves and other invertebrates that regularly constitute a part of the diet of these fish in Campeche [44].

A number of different methods have been employed globally for monitoring PSTs in bivalve mollusks. The traditional use of mouse bioassays (MBA) [45,46] has in recent years been replaced through regulations by instrumental chemical detection methods [47]. In North and South America, this is a post-column oxidation (PCOX) liquid chromatography with fluorescence detection (LC-FLD) method, whereas for European legislation, the formal reference method uses pre-column oxidation (Pre-COX) LC-FLD to quantify PSTs [48]. Both approaches have been subjected to validation, with advantages and disadvantages existing with each assay [49]. While other approaches based on receptor activity have also been validated and implemented into some monitoring programs [50], more recent developments have focused on the use of LC with tandem mass spectrometry (LC-MS/MS). One such method [51] utilizes a desalting clean-up step prior to acetonitrile dilution and analysis and has been subjected to both single-laboratory [52] and multi-laboratory validation [53].

In September and November 2016, eight marine sampling sites along the coast of the southeastern Gulf of Mexico were monitored for the presence of lipophilic and hydrophilic toxins [54,55]. Physicochemical parameters, inorganic nutrients and phytoplankton abundance were also determined. The aims of the study were to identify the toxin producer, evaluate physicochemical factors that may be associated with its proliferation and confirm the presence of PSTs in this dinoflagellate in the southeastern Gulf of Mexico.

## 2. Results

### 2.1. Physicochemical Variables

The variation of the water temperature (Table 1) was low in both months; in November, at station 8, it reached a maximum of 38.63 °C. Salinity showed high variation both between stations and between months, with a minimum value of 17 at station 2 in November and a maximum value of 41.57 at station 4 in September. The pH showed a maximum value of 9.63 at station 7 in September and a minimum value of 6.5 at station 6.3 in November. Dissolved oxygen (DO) showed a minimum value of 4.1 mg L^−1^ at station 7 in November.

In September, nitrites at all stations were above the maximum limit established for marine waters (MLEMW) [56]. As in November, a maximum value of 0.045 μmol L^−1^ was recorded at station 2. In September and November, nitrates were above the MLEMW. At stations 1, 2, 4 and 8, the ammonium concentrations were above the MLEMW. At all stations, orthophosphates were well below the MLEMW (Table 1).

### 2.2. Identification and Abundance of Pyrodinium bahamense

The cell width was measured above or below the cingulum, not taking into account the cingular lists: length without spines, including the apical horn, 45–65 (56 ± 5.81) µm, length with spines 52–95 (78 ± 10.76) µm, width 44–53 (53 ± 4.62) µm, length without spines/width ratio (L/W) 0.88–1.25 (1.06 ± 0.06). The apical horn was approximately 5 µm. Cells were without spines, or with only one (apical or antapical) spine or with both. Almost all the cells were solitary; some two-celled chains (attached daughter cells) were observed. Only one cell with slightly different proportions was found (length without spines 54 µm, length with spines 75 µm, width 46 µm, L/W 1.17) Figure 1 and Figure 2, Table 2.

In September, *P. bahamense* showed a maximum abundance of 3.13 × 10^4^ cells L^−1^ at station 7, followed by 1.90 × 10^4^ cells L^−1^ at station 2. At the other stations, the abundances ranged from undetected to 10^3^ cells L^−1^ in November (Figure 3).

### 2.3. Toxicity

#### 2.3.1. Lateral Flow Immunochromatographic Test (Rapid PSP Test)

Positive identification of PSTs by qualitative analysis (saxitoxin and analogs) for lateral flow immunochromatography (LFIC) was found in phytoplankton extracts. The intensity of the marked line was similar to a positive control (STX, FDA Reference Standard STD, was obtained from the US National Institute of Standards and Technology (NIST, RM 8642).

#### 2.3.2. HPLC-FLD Analysis

The toxin profile determined by post-column oxidation HPLC-FLD of *P. bahamense* in comparison to certified reference standards showed the presence of two analogs from the group of carbamoyl toxins (STX and NeoSTX; Figure 4). No sulfocarbamoyl and decarbamoyl toxins were detected in this study. The toxin content was quantified at 6.5 pg cell^−1^ (against external calibration standards. The molar percentages recorded for *P. bahamense* showed high molar percentages of carbamoyl toxins with 86.7% and 13.3% of STX and NeoSTX, respectively.

#### 2.3.3. UPLC-MS/MS Analysis

Figure 5 illustrates the MRM chromatograms obtained following HILIC-UHPLC-MS/MS analysis of STX and NeoSTX standards and the sample of *P. bahamense*. Peaks were observed for both STX and NeoSTX in the algal sample, with retention times identical to those determined in the calibration standard. Ion ratios determined between standard and sample were found to be similar (±10% deviation).

## 3. Discussion

The physicochemical variables (temperature, salinity, pH and DO) measured at the end of the rainy season and at the beginning of the windy season coincided with those reported by Poot-Delgado et al. [62], who concluded that their variation was influenced by the shallow depth in the study area (~1 m), recording higher values compared to the area more than 2.5 km offshore [63,64,65,66]. Similarly, the registered values of nitrogenous compounds are above the MLEMW, which could be influenced by the proximity of the study area to point sources of wastewater discharges resulting from various anthropogenic activities [67,68].

Phlips et al. [69] described, as physiological limits for *P. bahamense* in Florida, a temperature of 20 °C as the lower temperature limit and a tolerance to salinity between 10 and 45. The bloom potential of this dinoflagellate was most closely associated with shallow ecosystems with long water residence times, and peak biomass levels were correlated to nutrient concentrations in regions of high abundance.

In Mexico, *P. bahamense* has been reported from various states from the Mexican Pacific (Baja California, Baja California Sur, Guerrero, Oaxaca, Chiapas; see [32,35,59,70,71,72,73]) and in the Gulf of Mexico from the states of Campeche and Yucatan [60,61]. Although several publications have been documented with light micrographs [32,35,59,70,71,72,73,74], SEM micrographs are still rare: [59,74]—vegetative cells, [72]—cysts.

In the present study, we followed [33], who found no significant morphological differences between the two varieties, var. *bahamense* and var. *compressum* [37]. Mertens [11] suggested that instead, they are probably different life stages and that *P. bahamense* is a species complex represented by two clearly separated ribotypes, Indo-Pacific and Atlantic–Caribbean ones.

Cusick and Duran [75] reported the existence of genotypes in natural *P. bahamense* subpopulations in Florida and Puerto Rico, with relevant insights on underlying genetic factors influencing the potential for toxin variability among natural subpopulations of HAB species.

The toxicity profile of *P. bahamense* is relatively simple, with most isolates from Indo-Pacific producing only dc-STX, STX, NeoSTX, B1 and B2 toxins [7]. This toxin profile is typical of both natural environmental samples and cultures [76,77,78,79] (Table 3).

Nevertheless, the isolates from Guatemala contained STX, neoSTX, B1, GTX2, GTX3 and GTX4 [81] (Table 3); in Florida, *P bahamense* most likely produced STX, dc-STX and B1 as detected in puffer fish and HAB of *P. bahamense* [22] (Table 3), and one strain from Isla San José in the Gulf of California produced only STX [72] (Table 3).

Studies on bivalves contaminated by *P. bahamense* in Maylasia and the Phiippines showed that there were few differences in toxin profiles of dinoflagellate and shellfish [7,82,83].

Chromatographic profiles have been determined by HPLC-FLD and UHPLC-MS/MS from shellfish sampled during the main bloom of *P. bahamense*, which had the greatest impact on public health in Mexico [14,55]. A concentration of 7396 μg eq STX kg^−1^ was determined in the rock oyster *Crassostrea iridescens* (Hanley) from the Bay of Acapulco, Guerrero, during the 1995 HAB event; between 8889.3 and 32,272.5 μg eq STX kg^−1^ occurred in the clam *Donax gracilis* Hanley from Zapotal and 14,445 μg eq STX kg^−1^ in the mangrove mussels *Mytella strigata* (Hanley) from Puerto Madero, both in Chiapas, during the HAB of 2001 (regulatory limit for PSTs in flesh is 800 µg saxitoxin equivalents (STX eq.) per kg of shellfish flesh). The profiles by HPLC-FLD were STX, GTX2, dcSTX, dcGTX2, dcGTX3 and B1 in *C. iridescens*, STX, NeoSTX, GTX2, GTX3 and B1 in *D. gracilis* and STX, GTX2, GTX3 and B1 in *M. strigata*. In Salina Cruz, Oaxaca, in 2002, 306.3 μg eq STX kg^−1^ was still detected in the spiny oyster *Spondylus calcifer* Carpenter, with only STX present. Additionally, UHPLC-MS/MS analysis was used to detect GTX1, GTX5, GTX6, doSTX, dcNEO and C1 [14,55]. The two PST detection methods in this study are both thoroughly validated and have been found to generally compare well when applied to shellfish samples. A recent study assessing the use of six detection methods for analysis of shellfish from Latin America, including Mexico, showed good correlations between all instrumental methods applied. The HPLC-FLD method has been previously validated for a wide range of PST analogs, although the LC-MS/MS has the further advantage of incorporating a larger number overall. Both methods are, however, well suited to the accurate quantitation of individual PST analogs and the summation of PSTs to estimate total toxin levels.

Puffer fish (Tetraodontidae) from southeastern Mexico can also be vectors of PSTs associated with HABs of *P. bahamense* and could be the cause of SPFP in human cases [44,55] reported in the Yucatan Peninsula [85]. The profiles of sodium channel blocker neurotoxins (TTX and analogs + STX and analogs) in five species of puffer fish of the genus *Sphoeoroides* and *Lagocephalus* were recently evaluated along the central coast of Campeche, are attributed to *P. bahamense*, and were composed of PSTs (STX, NeoSTX, GTX2, GTX3, dcGTX2 and dcGTX3 [42] and TTX analogs. The presence of TTX and STX analogs was also confirmed by UHPLC-MS/MS, detecting TTX, 4-epiTTX, MonodeoxyTTX, 4,9-anhydroTTX and 11-norTTX-6(R)-ol as well as STX and dcSTX [54,55].

This was observed by Núñez-Vázquez et al. [42,54] indirectly in puffer fish, probably due to the toxins transmitted via bivalves and other invertebrates that regularly make up part of the diet of these fish in Campeche coastal waters [44]. Previous studies by [22,86] demonstrated the presence of high concentrations of PSTs in puffer fish from Florida, resulting from the *P bahamense* HAB event.

PSP has been the main public health problem due to bivalves in Mexico [14,28,55]. The HABs of *P. bahamense* have caused 92.78% of PSP cases, with 476 human intoxications, all occurring along the southeastern Mexican Pacific coast [55].

Studies that confirm the presence of PSTs in natural samples are scarce, and few reports are available that have confirmed toxin-producing microalgae, either from environmental samples or laboratory cultures (Table 3) worldwide with this species. This study, in addition to generating data on the microalgal toxin profile, has allowed for comparison with those toxin profiles from other regions of the world and from other species of dinoflagellates (e.g., *Alexandrium* spp. and *Gymnodinium catenatum*) that produce PSTs. *Pyrodinium bahamense* is one of the harmful microalga that has caused the highest number of human poisonings and deaths from PSP at global and regional levels (8). To the best of our knowledge, this is only the second time that the production of PSTs by *P. bahamense* has been confirmed in the Atlantic of the American continent, and worldwide, it is one of the few cases of toxins confirmation in this species. As such, this study contributes to the evaluation of possible risks to public and animal health in the area, as well as socioeconomic impacts related to fisheries and aquaculture industries. Additionally, the knowledge of the toxin profile of *P. bahamense* has allowed for confirmation of the possible etiology of these toxins that can be transferred to other trophic levels, including for example, puffer fish that are consumed and have caused human poisonings in this region.

## 4. Conclusions

The occurrence of the gonyaulacoid dinoflagellate *P. bahamense* in the southeastern Gulf of Mexico using light microscopy and SEM was related to elevated nutrient content at the sites of rainwater discharge or those near anthropogenic activities. STX and NeoSTX were detected by chromatographic techniques (IFL, HPLC-FLD and UHPLC-MS/MS). This study is the first report that confirms the presence of PST in *P. bahamense* in Mexican waters of the Gulf of Mexico.

## 5. Materials and Methods

### 5.1. Seawater Samples

In September and November 2016, seawater samples were taken at eight stations (site depth ca. 1 m) located along the central coast of the state of Campeche. All sampling sites were influenced by rainwater discharge or were near the sites with anthropogenic activities (Figure 6).

### 5.2. Phytoplankton Sampling and Nutrient Determination

At each site, surface seawater samples were collected with a 1 L plastic bottle; a 100 mL aliquot was used to determine cell abundances of phytoplankton taxa. Immediately after collection, samples were fixed with a neutral Lugol solution and subsequently preserved by adding 37% neutralized formalin to a final concentration of 4% [87]. Cells were counted after sedimentation in a 10 mL cylinder following Reguera et al. [88]. Additionally, horizontal tows were taken with a 20 μm mesh conical hand net. The material collected for the *P. bahamense* analysis was placed in glass vials and fixed using the same procedure as for the cell counting. Net samples were also examined under a light microscope for accurate identification of *P. bahamense*. In situ water temperature (°C), salinity, pH and DO were measured using a using a Hanna multiparameter probe, model HI9828, equipped with a sensor, model HI769828, and a Hach multiparameter probe, model HQ40d (Hanna Instruments Inc., Woonsocket, RI, USA). Orthophosphate, ammonium, nitrite, nitrate and silicate analyses were performed in the laboratory following [89].

*Pyrodinium bahamense* cells were quantified according to the Utermöhl technique [90] using an inverted Zeigen microscope (Xiamen, China) equipped with phase-contrast objectives 10x/0.25 Ph1 ADL and LD 25x/0.30 Ph1. The cell abundance was expressed in cells L^−1^.

To make the thecal plate arrangement of *P. bahamense* visible, selected cells were stained with Calcofluor White M2R [91] by adding a drop of a 0.2% stain in aqueous solution to a water mount. Cells were observed under epifluorescence microscopy (Axio Scope.A1, Carl Zeiss, Oberkochen, Germany) with filter set 18 shift free EX BP 390–420 (excitation), BS FT 425 (optical divider) and EM LP 450 (emission), using the Plan-Neofluar 40×/0.75 and 63×/0.95 Korr objectives (total magnification 400× and 630×, respectively). Photomicrographs were taken with a Carl Zeiss Axiocam 506 color camera (6 MP) using the ZEN 2012 SP2 program (Carl Zeiss Microscopy GmbH, Göttingen, Germany).

### 5.3. Identification of Pyrodinium bahamense

Cell measurements (*n* = 50) were made using the 40×/0.65 objective and an Olympus BX51 compound microscope. Four samples were examined in a JEOL JSM-7600F scanning electron microscope (SEM) at a working distance of 15 to 21 mm and a voltage of 1.2 to 5.0 kV after a preliminary wash in distilled water, followed by dehydration in a series of ethanol solutions of increasing concentration (30, 50, 70, 90 and 100%), air drying on 0.5″ aluminum mounts and sputter coating with gold–palladium using a Polaron SC7640 High Resolution Sputter Coater (Quorum Technologies, Newhaven, UK).

### 5.4. Toxicity

#### 5.4.1. Lateral Flow Immunochromatography Test (Rapid PSP Test)

Analysis for paralytic shellfish toxins (saxitoxin and analogs) were conducted by Lateral flow immunochromatography (LFIC) using rapid test kits provided by Scotia Rapid Testing (Scotia Rapid Testing Ltd., Nova Scotia, Canada), a qualitative lateral flow test for detection of PST in shellfish and phytoplankton for PSP (SRT PSP). The development and validation of the SRT PSP kit using the standard protocol is described by Scotia test [92].

#### 5.4.2. HPLC-FLD Analysis

Phytoplankton samples from station 7 were harvested by filtration through glass fibers (GF/F Whatman) and immediately frozen at −20 °C. These filters were used for toxin analysis. PSTs were analyzed by high-performance liquid chromatography and fluorescence detection (HPLC-FLD) [77] modified by [93]. PSTs were extracted by adding 2 mL of acetic acid (0.03 N) to each *P. bahamense* sample containing the filter and sonicated three times for 5 min each. The extract was centrifuged for 15 min at 14,000 rpm (HERMLE Z 216 microcentrifuge, Labortechnik GmbH, Wasserburg, Germany), and the supernatant was filtered through 0.45 μm filters using syringe filters of glass fiber (PVDF Millex membrane, 25 mm diameter). For the derivatization, 150 μL of each extract was mixed with 37 μL of hydrochloric acid 1 M and heated for 15 min at 90 °C. Cool to room temperature, after adding 75 μL of 1 N sodium acetate. The extracts were injected in the liquid chromatograph in separate runs for the identification and quantification of the PSTs (standards of saxitoxin (STX), neosaxitoxin (NeoSTX), goniautoxin-1,4 (GTX 1,4), decarbamoylsaxytoxin (dcSTX), decarbamoylgoniautoxin-2,3 (dc GTX 2,3), and N-sulfocarbamoyl-11-hydrosulfate (C1,2).

The chromatography system used was an HP 1100 (Agilent Technologies, Santa Clara, CA, USA) consisting of an autosampler, degasser, quaternary pump, two binary pumps used for post-column reactions, a fluorescence detector, a C-18 column, and a post-column reactor. The PSTs were detected using an excitation wavelength of 333 nm and an emission wavelength of 390 nm. Two criteria were used for the identification of the PSTs: the retention times and co-elution with commercial standards of (STX), (NeoSTX) (National Research Council Canada, Institute for Marine Biosciences, Certified Reference Material Program, Halifax, NS, Canada).

#### 5.4.3. UPLC-MS/MS

A Waters (Manchester, UK) Acquity UPLC I-Class coupled to a Waters Xevo TQ-S tandem quadrupole mass spectrometer (MS/MS) was used for UHPLC-MS/MS analysis. A Waters 1.7 µm, 2.1 × 150 mm Waters Acquity BEH Amide UPLC column held at +60 °C was used for chromatographic separation of toxin analogs in conjunction with a Waters VanGuard BEH Amide guard cartridge. Cleaned-up sample extracts were injected onto the UHPLC with an injection volume of 2 µL. Mobile phases and UHPLC gradient conditions were exactly as described by [52]. The MS/MS tune parameters were as follows: 150 °C source temperature, 600 °C desolvation temperature, 1000 L h^−1^ desolvation gas flow, 7.0 Bar nebulizer gas flow, 150 L h^−1^ cone gas flow and 0.15 mL min^−1^ collision gas flow. Capillary voltage was held at 0.5 and 2.5 kV for positive and negative ionization modes, respectively. Multiple Reaction Monitoring (MRM) transitions were exactly those described by [52,94]. Quantitation of toxins was performed against toxin standards available as certified reference standards. Five additional analogs (C3, C4, dcGTX1, dcGTX4 and GTX6) were incorporated into the method, with quantitation performed using the calibrations generated from their nearest structural analog, using experimentally determined relative response factors [51]). Toxicity equivalence factors (TEFs) were taken from EFSA recommendations where feasible [94], with others used as described by [94]. The method has been validated in shellfish internationally through collaborative study with well defined method performance characteristics [53]. Assessment of the method when applied to water or algae samples has previously demonstrated the absence of any significant matrix effects.

## Figures and Tables

**Figure 1 toxins-14-00760-f001:**
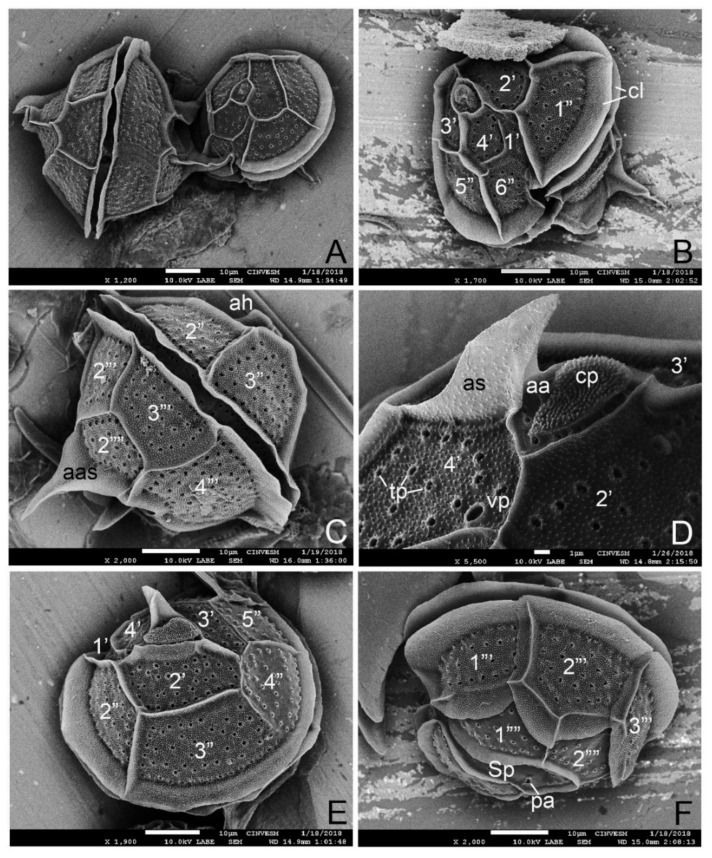
Thecal morphology of *Pyrodinium bahamense* (viewed with scanning electron microscopy). (**A**) two cells in dorsal (left) and apical–ventral left-side view; (**B**) cell in apical–ventral left-side view; (**C**) cell in dorsal left-side view; (**D**) a fragment of the epitheca with the apical pore complex and the apical spine; (**E**) cell in apical–dorsal left-side view; (**F**) cell in antapical left-side view. Plate labels: 1′–4′, the apical plates; 1″–6″, the precingular plates; 1‴–5‴, the postcingular plates; 1″″ and 2″″, the antapical plates; aa—the anterior attachment pore, aas—the antapical spine, ah—the apical horn, as—the apical spine, cl—the cingular lists, cp—the canopy (also known as the cover plate or the closing plate), pa—the posterior attachment pore, Sp—the posterior sulcal plate, tp—the trichocyst pores, vp—the ventral pore. The plates are named mainly according to Balech [33]. Scale bars: 10 µm in (**A**–**C**,**E**,**F**), 1 µm in (**D**).

**Figure 2 toxins-14-00760-f002:**
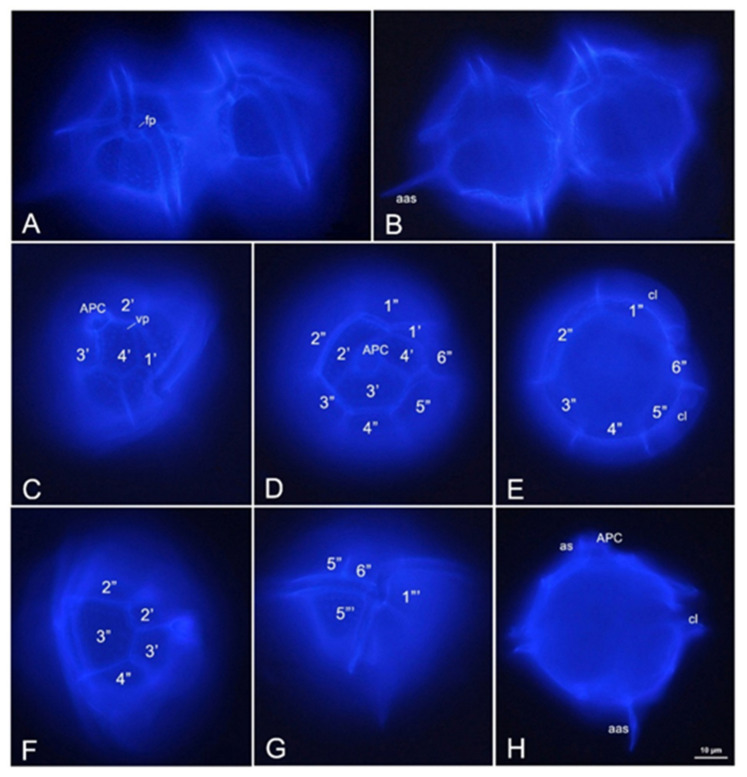
Thecal morphology of *Pyrodinium bahamense* (viewed with epifluorescence microscopy). (**A**,**B**) two cells in ventral view on different focal planes; (**C**) apical–ventral view; (**D**,**E**) apical view on different focal planes; (**F**) dorsal–apical view; (**G**,**H**) ventral view on different focal planes. Plate labels: 1′–4′, the apical plates; 1″–6″, the precingular plates; 1‴–5‴, the postcingular plates; Xaa—the anterior attachment pore, aas—the antapical spine, APC—the apical pore complex, as—the apical spine, cl—the cingular lists, vp—the ventral pore. Scale bar: 10 µm.

**Figure 3 toxins-14-00760-f003:**
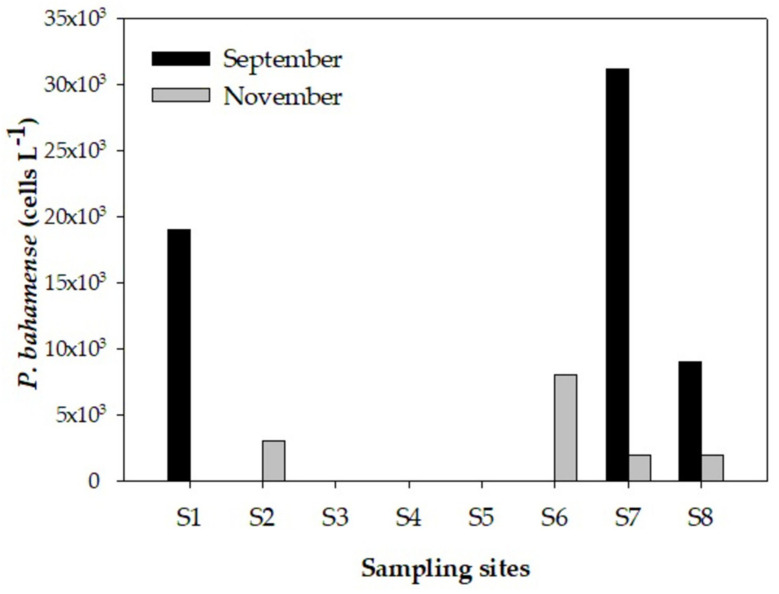
Temporal variation of *P. bahamense* at eight stations (S1–S8) along the Campeche central coast, southeastern Gulf of Mexico.

**Figure 4 toxins-14-00760-f004:**
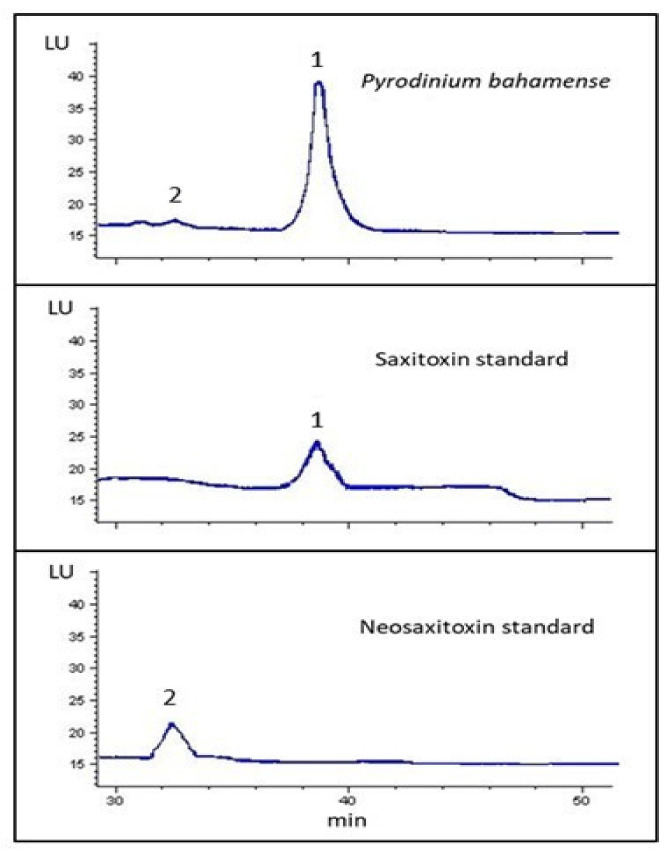
Post-column oxidation HPLC-FLD chromatograms showing the presence of STX (1) and NeoSTX (2) in *P. bahamense* from the coastal waters of Campeche.

**Figure 5 toxins-14-00760-f005:**
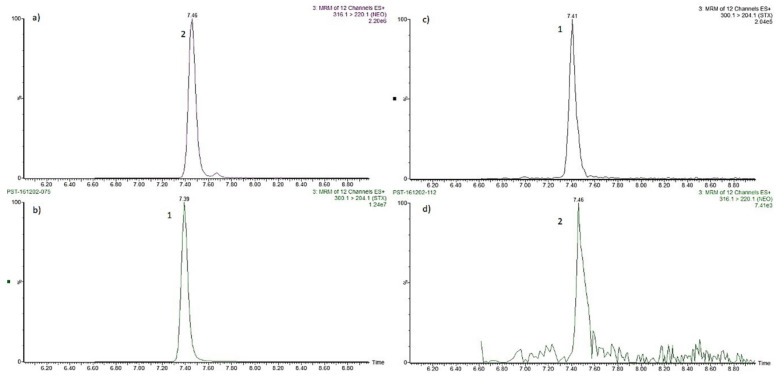
Confirmation of PSTs in *P. bahamense* from the coastal waters of Campeche for UPLC-MS/MS. (**a**) NeoSTX in calibrant, (**b**) STX in calibrant, (**c**) STX in sample, (**d**) NeoSTX in sample.

**Figure 6 toxins-14-00760-f006:**
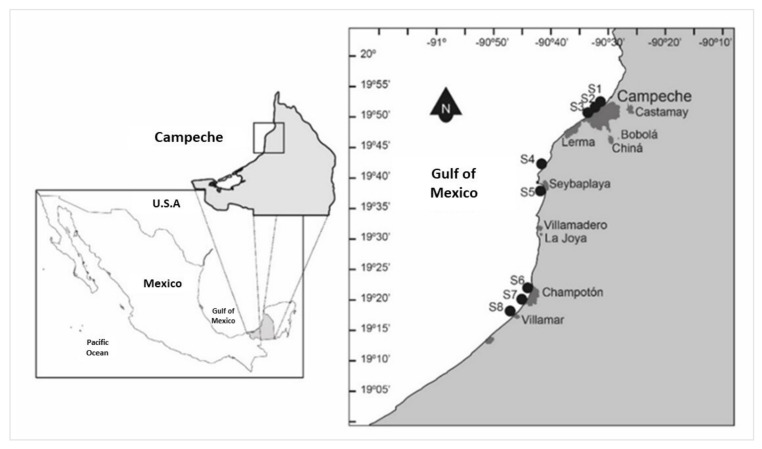
Location of study area and sampling sites (S1–S8) in the coastal waters of the state of Campeche.

**Table 1 toxins-14-00760-t001:** Variability of physicochemical characteristics from eight sampling sites along the central coast of Campeche, southeastern Gulf of Mexico. * Water quality criteria for the protection of aquatic life [56].

**Physicochemical Variables**								
	**T (°C)**		**Salinity**		**pH**		**DO (mg L^−1^)**	
Stations	September	November	September	November	September	November	September	November
S1	31	27.5	37.8	24	9.29	7.06	6.2	6.9
S2	31	28.6	39.95	17	9.26	6.9	6.7	4.12
S3	31.37	28.5	40.54	21	9.05	6.5	7	5.7
S4	31.31	28.8	41.57	39	9.37	7.7	7	7.45
S5	31.79	29.4	40.01	30	9.29	6.6	7.3	7.53
S6	32.06	29.4	28.24	18	9.14	7	7.5	6.53
S7	32.8	30.1	38.42	24	9.63	9	7.9	4.1
S8	32.31	38.63	39.47	35	9.29	7.3	7.7	6.79
Mean	31.71	30.12	38.25	26	9.29	7.26	7.16	6.14
SD	0.65	3.52	4.21	7.96	0.17	0.79	0.55	1.37
**Nutrients (µmol L^−1^)**								
	**Nitrites**		**Nitrates**		**Ammonium**		**Orthophosphates**	
Stations	September	November	September	November	September	November	September	November
S1	0.034	0.021	0.196	0.194	0.219	0.141	0.054	0.008
S2	0.062	0.045	0.249	0.83	0.273	0.235	0.059	0.04
S3	0.028	0.03	0.182	0.457	0.074	0.105	0.046	0.01
S4	0.012	0.024	0.098	0.389	0.168	0.145	0.085	0.011
S5	0.006	0.021	0.07	0.29	0.073	0.056	0.026	0.001
S6	0.01	0.032	0.272	0.845	0.093	0.046	0.012	0.02
S7	0.011	0.006	0.134	0.462	0.095	0.046	0.05	0.006
S8	0.021	0.013	0.082	0.263	0.138	0.088	0.063	0.021
Mean	0.023	0.024	0.160	0.466	0.142	0.108	0.049	0.015
SD	0.018	0.012	0.076	0.247	0.073	0.065	0.022	0.012
* Upper limits established for marine coastal waters	0.002		0.04		0.01		5	

**Table 2 toxins-14-00760-t002:** Vegetative cell measurements of *Pyrodinium bahamense* from the Mexican Pacific, the Greater Caribbean and Central America.

Length (Cell Body): Range or Average, μm	Width: Range or Average, μm	Locality	Reference
50	48	New Providence Island, the Bahamas	[1]
66 (86 with spines)	54	Gulf of Tehuantepec, east coast, and Pacific off El Salvador	[57]
33.8–77.1 (42.5 ± 10.8)	33.8–67.7 (39.9 ± 7.8)	Tampa Bay, West Florida	[37]
33–47	47–53	Gulf of Nicoya, Costa Rica	[34]
33–77	37–67	Douglas Cay and Twin Cay (coral-reef mangrove lagoons), Belize	[58]
41.9	43.8	Isla San José, Gulf of California	[59]
43 (without the apical horn)	41.5	Bahía de Campeche, Gulf of Mexico	[60]
33–47	37–52	Baja California Peninsula, southern coast	[35]
39–59 (48.6 ± 5.0); 48–90 (65.9 ± 11.1) with spines	40–50 (45.3 ± 2.7)	Marinas along the northern Yucatan Peninsula, Gulf of Mexico	[61]
30–52	35–55	Mexican Pacific	[32]
27.30–81.88 (46.01 ± 9.35)	22.60–83.34 (48.11 ± 8.97)	Caribbean, Florida	[11]

**Table 3 toxins-14-00760-t003:** Toxin content and PST profile of natural samples and cultures of *Pyrodinium bahamense*.

Locality, Year	Profile	Toxin Content (pg STX per Cell)	Reference
Palau, 1980	Natural: STX, NeoSTX, GTX5	1.5–1.4 × 10^−4^ MU/cell *	[80]
Guatemala, 1987	Natural: STX, NeoSTX, B1, GTX2, GTX3 and GTX4	-----	[79,81]
Sabah, Malaysia, 1991	Culture: STX, NeoSTX, GTX5, GTX6 and dcSTX	0.66–3.98	[82]
Masinloc Bay, Philippines, 2000 and 2002	Natural: STX, NeoSTX, B1 Culture: STX, NeoSTX, dcSTX, B1 and B2	0.66–5.35 0.54–1.33	[83]
Indian River Lagoon, Florida, U.S., 2005	Natural: STX, B1 and dcSTX Culture: STX and B1	3.28 2.02–12.74	[22]
Isla San José, Gulf of California, 2008	Culture: STX	0.31	[72]
Lagoon, Red Sea, 2013–2014	Culture: STX	2	[84]
Campeche, southeastern Gulf of Mexico, 2016	Natural: STX and NeoSTX	6.5	This study

* 1 MU (mouse unit) = 0.16 µg STX eq (equivalent).

## Data Availability

Not applicable.

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
