# Peer review of "Paralytic Shellfish Toxins of Pyrodinium bahamense (Dinophyceae) in the Southeastern Gulf of Mexico"

_toxins, 2022, doi:10.3390/toxins14110760_

Round 1

Reviewer 1 Report

Dear Authors and Editor,

the msc "Paralytic shellfish toxins of Pyrodinium bahamense in the south eastern Gulf of Mexico" (toxins-1878407) describes a small data set on Pyrodinium and associated phyto community and environmental conditions from a particular region.

It is clear and well written, though not an English native. Some very minor details could be pointed out but overall, the reading is very straightforward. However, some major revisions should be carried out mostly related with the scope of the work, a course has to be set in my opinion. It is paramount to decide what to include and detail in the discussion for example. As the authors will see through the comments, many things were found to be a bit out of scope or poorly detailed or discussed to the detriment of others.

Keep the work as a full paper but then a “big loop” has to be given to what information is presented (the environmental information as it is discussed adds very little to the work) or move to a short paper where the focus is the detection of Pyrodinium in the region and associated toxicity and inferences on regional monitoring programs, shellfish safety information, EWS, etc

Research on ecology, bloom dynamics and toxicity are always relevant and worth to be published. It is always information that consolidates the time series that exist for the region, consequently a more accurate modelling and more robust predictions of occurrence and impact, the basis for the implementation of an effective warning system for food safety and coastal management if necessary.

Extended comments on the msc are in the document attached 

Reviewer 2 Report

This manuscript reports the detection of PSP in the coast of the southeastern Gulf of Mexico and P. bahamene as the PSP-causing species. It is the first report on PSP in the studied regions; however, as mentioned Introduction, the P. bahamense is common in the Gulf of Mexico and it can be present elsewhere in that area. So, it is not special. In addition, I found that most data used in the MS were obtained by one time and simple monitoring. Thus, the amount of data used in the MS is small, and the reliability of the data analyzed in statistics may be lowered. Considering these, I feel that the scientific value of this MS is little, and cannot encourage it to be published in journal.

Specific

-Table 2. Indicate specific appearance information for 8 sites.

-2.3 subtitle and Table 4. Statistical analysis is meaningless because the number of data is small.

-2.4.1 Subtitle, It should be moved to M&M, not results.

-Fig. 7. Did you predict the chemical structure by yourself NMR? or just bring other source? Clarify these. If just bring others, it is not necessary in this MS.

-L192, 197, P. bahamense, italic

-L 201-234, Authors monitored one times at 8 sites, and discussed too much. Describe shortly to connect your results.

-L236-242, Why did not analyze DNA for discrimination of var. bahamense and var. compressum? In addition, is it same species or genotype of P. bahamense?

Reviewer 3 Report

The present manuscript entitled "Paralytic shellfish toxins of Pyrodinium bahamense in the southeastern Gulf of Mexico" (toxins-1878407) is written correctly and has a good structure; moreover, it has all the necessary parts. The article is interesting from an analytical and environmental point of view; therefore, it should interest the reader. I proposed improvements in method description and with a presentation of figures. The paper meets Toxins' requirements, and I recommend the article for publication in Toxins following the common editing stage. My current decision is a major revision.

I hope that the tips and observations included in the comments will help you decide regarding the publication in Toxins. More specific comments and observations are presented below.

1. The citation should be written in the correct order. It can be found a record, e.g., [9,10,7,11], and it should look like this [7,9-11]. I am asking for improvement in all places.

2. Page 2, first paragraph; page 17, line 362. Web pages and the access date should be numbered and written in the literature.

3. Introduction. A paragraph on analytical techniques used for toxin analysis may be added. It would be interesting for the reader.

4. Page 2, lines from 81to 89; and page 4, lines 104-109. This text is too monotonous and needs to be edited.

5. Figure 2. The title with the unit in parentheses should appear on the y-axis.

6. Drawings are sometimes written in lowercase and sometimes in uppercase. This should be standardized along with the formatting of all drawings.

7. Tables 2, 3, 4, and Figures 3, 4, 5, and 8 should be better commented in the text in appropriate places. Figures 6 and 7 should be mentioned and discussed in the text.

8. Please change “relationship” to “relation”. "Relationship" is more about people.

9. Figures with chromatograms. The axes should contain a title along with the unit. The values should be clearly visible. The redundant text should be deleted. Is it possible to export the data and prepare the chromatograms in better quality?

10. Chromatographic methods. Is it possible to perform validation and determine the basic validation parameters?

11. Page 10, lines 183 and 184. Are the authors sure that it is Table 6 here?

12. Page 10, line 184. How does external calibration relate to interference effects? What can be done in the event of strong interference effects? How would you deal with them? What types of interference effects could occur?

13. Page 13. The use of commas for numerical values is problematic and confusing.

14. Conclusions. This part should be expanded. Please, emphasize clearly the advantages of the research carried out.

15. Does the developed methods have disadvantages? What are the limitations?

16. What are the plans for the future?

17. Page 15, lines 305 and 307. Figure 9 should be instead of Figure 8.

18. Page 17, line 372. 1.0 M should be instead of 1.0 N.

19. The notation of units should be standardized throughout the manuscript.

20. Page 17, line 399. The method from [94] should be described in detail here.

21. Page 17, line 408. Boundy et al., 2015 ?

22. References. Please check with journal requirements. Several typos may be found.

I hope that the comments presented will help improve the article.

Author Response

Consulte el archivo adjunto

Round 2

Reviewer 3 Report

Dear Authors,

Thank you for your meticulous consideration of my comments. The paper has improved substantially and, in my opinion, is suitable for publication after minor revision.

1. Conclusions. Please expand this part 2 times, adding some important points with conclusions.

2. Page 13, line 297. There is a mistake in [889].

3. Page 13, line 305. There is a mistake in [8980].

4. References. Please add the access date to the Web pages.

Please check other typos and possibly improve the quality of some drawings.

Round 3

Reviewer 1 Report

Dear Authors

congratulations on the reformulation of the manuscript, it is much more focused and appealing. I don't have anything else to add, for my part, the msc can advance for publication in the current version.
